# Metabolic Activation and DNA Interactions of Carcinogenic *N*-Nitrosamines to Which Humans Are Commonly Exposed

**DOI:** 10.3390/ijms23094559

**Published:** 2022-04-20

**Authors:** Yupeng Li, Stephen S. Hecht

**Affiliations:** 1Masonic Cancer Center, University of Minnesota, Minneapolis, MN 55455, USA; hecht002@umn.edu; 2Department of Medicinal Chemistry, College of Pharmacy, University of Minnesota, Minneapolis, MN 55455, USA

**Keywords:** *N*-nitrosamines, metabolism, DNA adducts, NDMA, P450s

## Abstract

Carcinogenic *N*-nitrosamine contamination in certain drugs has recently caused great concern and the attention of regulatory agencies. These carcinogens—widely detectable in relatively low levels in food, water, cosmetics, and drugs—are well-established and powerful animal carcinogens. The electrophiles resulting from the cytochrome P450-mediated metabolism of *N*-nitrosamines can readily react with DNA and form covalent addition products (DNA adducts) that play a central role in carcinogenesis if not repaired. In this review, we aim to provide a comprehensive and updated review of progress on the metabolic activation and DNA interactions of 10 carcinogenic *N*-nitrosamines to which humans are commonly exposed. Certain DNA adducts such as *O*^6^-methylguanine with established miscoding properties play central roles in the cancer induction process, whereas others have been linked to the high incidence of certain types of cancers. We hope the data summarized here will help researchers gain a better understanding of the bioactivation and DNA interactions of these 10 carcinogenic *N*-nitrosamines and facilitate further research on their toxicologic and carcinogenic properties.

## 1. Introduction

Peter Magee and John Barnes reported in 1956 that *N*-nitrosodimethylamine **1** (NDMA, Figure 1), a simple water-soluble compound with only 11 atoms, readily induced liver tumors in rats [1]. This was remarkable because most carcinogenesis studies at the time had been performed with higher molecular weight non-water-soluble compounds such as polycyclic aromatic hydrocarbons. Sakshaug et al., and Ender and Ceh made the connection between NDMA formation from nitrite-treated herring meal and liver toxicity in farm animals and provided evidence for the occurrence of this carcinogen in smoked fish and meat [2,3]. Thus, concern arose that *N*-nitrosamines in food treated with nitrite could be a carcinogenic hazard to humans. Sen and colleagues demonstrated the presence of *N*-nitrosamines in cured meat products [4] and this was followed by a surge in interest in the *N*-nitrosamine contamination of foods, which persists to the present.

In the meantime, cancer researchers investigated the powerful carcinogenic properties of multiple structurally diverse *N*-nitrosamines. Druckrey and co-workers and Lijinsky and colleagues demonstrated the carcinogenicity and frequent organoselectivity of multiple *N*-nitrosamines [5,6]. A review published in 1984 by Preussmann and Stewart summarizes the carcinogenicity of more than 200 *N*-nitrosamines, which commonly affect specific organs in laboratory animals, independent of the route of administration [7]. A book by Lijinsky also summarizes the extensive carcinogenicity data [5]. Bogovski and Bogovski published a summary of the carcinogenic activity of nitroso compounds in different animal species; NDMA induced tumors in 16 different animal species and NDEA in 26 ranging from rainbow trout to cynomolgus monkey [8]. Such a remarkable database hardly exists for any other type of carcinogen.

Thus, there was intense interest in the possible role of *N*-nitrosamines in human cancer in the latter part of the 20th century. The International Agency for Research on Cancer (IARC) held a series of regular meetings dedicated to this subject between 1966 and 1991. Figure 2 shows the participants at the 1983 meeting in Banff, Canada, a clear indication of the high interest in the topic. Fortunately, methods were developed to decrease levels of *N*-nitrosamines in food, beer, and other consumer products and interest in the topic waned somewhat in the early part of this century. Recently, however, concern regarding *N*-nitrosamine contamination of consumer products has re-emerged as they were found in certain pharmaceutical agents and drinking water [9]. It is worth noting that the carcinogenic potency of most *N*-nitrosamines is so great that they are excluded from the widely used Threshold of Toxicological Concern concept in the risk assessment of exposure to potential carcinogens in food and other consumer products [10].

All *N*-nitrosamines require metabolism to exert their carcinogenic properties. The electrophiles produced in these simple metabolic pathways, generally catalyzed by cytochrome P450 enzymes, readily alkylate DNA initiating the carcinogenic process. These critical pathways are the subject of this review of the 10 *N*-nitrosamines illustrated in Figure 1.

## 2. Overview of Carcinogenic *N*-Nitrosamines to Which Humans Are Commonly Exposed

*N*-Nitrosamines are the products of nitrosation reactions occurring on the *N* atoms of secondary and tertiary amines. They can be formed during water and food processing, tobacco curing, and drug and cosmetics manufacturing; they can also be formed endogenously. The compounds shown in Figure 1 represent an important family of carcinogens that are closely related to our daily lives [9,11]. 

### 2.1. Carcinogenic N-Nitrosamines Occurring in Food

The total *N*-nitrosamines occurring in food was estimated to be an average of 6.7 ± 0.8 ng/g, ranging from 0 to 120.8 ng/g [9]. NDEA was most frequently detected in 387 samples of agricultural food, whereas NDMA occurred at the highest concentration in seasoning, especially in processed fish (12.6–322.9 ng/g) and some oils (>10 ng/g) [12]. The average estimated concentrations of some common *N*-nitrosamines detected in food follows the descending order as NDMA (2.2 ± 0.3 ng/g), NDBA (1.5 ± 0.5 ng/g), NPYR (1.5 ± 0.2 ng/g), NDEA (0.9 ± 0.3 ng/g), NPIP (0.5 ± 0.1 ng/g), NMOR (0.05 ± 0.01 ng/g), NMEA (0.04 ± 0.01 ng/g), and NDPA (0.02 ± 0.01 ng/g) [9]. 

### 2.2. Carcinogenic N-Nitrosamines Occurring in Water

NDMA has been detected in potable water and gave rise to regulatory concerns regarding its presence in drinking water. The occurrence of NDMA is considered to be due to chloramination during water disinfection [13,14]. NDMA is the most prevalent *N*-nitrosamine contaminant in drinking water (average: 17.7 ± 4.7 ng/L), accounting for 5–13% of the total observed *N*-nitrosamines (average: 39.4 ± 10.5 ng/L; range: 2.8–309.0 ng/L) in water. The other carcinogenic *N*-nitrosamines detected in potable water include NPIP (7.9 ± 4.0 ng/L), NPYR (5.5 ± 2.6 ng/L), NDEA (4.2 ± 0.8 ng/L), NDBA (1.7 ± 0.6 ng/L), NMOR (0.9 ± 0.2 ng/L), NMEA (0.6 ± 0.1 ng/L), and NDPA (0.4 ± 0.03 ng/L) [9].

### 2.3. Carcinogenic N-Nitrosamines Occurring in Drugs

Drug contamination with *N*-nitrosamines has attracted great attention due to the recent recalls of more than 1400 lots of drugs from the U.S. market [15]. The presence of NDMA has been confirmed by the U.S. Food and Drug Administration (FDA) in some batches of ranitidine, nizatidine, metformin, and valsartan since 2018 [16]. Similarly, the detection of NDEA has caused the withdrawal of some types of irbesartan, losartan, and valsartan from the market [16]. The discovery of NDMA and NDEA has led regulatory agencies to conduct further analysis of the *N*-nitrosamine impurities in the affected drug products [17]. The U.S. FDA has identified 5 *N*-nitrosamines that were detected in drugs; they are NDMA, NDEA, *N*-nitroso-*N*-methyl-4-aminobutanoic acid (NMBA), *N*-nitroso-isopropylethylamine (NIPEA), and *N*-nitrodiisopropylamine (NDIPA). Two *N*-nitrosamines—NDBA and *N*-nitrosomethylphenylamine (NMPA)—are also considered to be theoretically present in drug products. The FDA-recommended acceptable intake limits for the carcinogenic *N*-nitrosamines discussed in this review are 96 ng/day (NDMA) and 26.5 ng/day (NDEA) [17].

### 2.4. Carcinogenic N-Nitrosamines Occurring in Cosmetics

The total *N*-nitrosamines occurring in cosmetics was estimated to be 1507 ± 752 ng/g, ranging from 0 to 49,000 ng/g. The relatively high concentrations of *N*-nitrosamines observed in cosmetics including hair care products, soaps, shampoos, lotions, and others are predominantly contributed by NDELA. It accounted for 99% of total *N*-nitrosamines in these products. The other minor *N*-nitrosamine contaminants are NMOR (~0.99%) and NDMA (~0.01%) [9]. NDEA has also been detected in cosmetics at levels ranging from 0–40.9 ng/g [18]. NDELA is formed by nitrosation of triethanolamine and diethanolamine [19] since they are readily present in cosmetics together with the nitrosating agents such as nitrite [18]. 

## 3. Metabolic Activation and DNA Interactions of Carcinogenic Acyclic *N*-Nitrosamines

As shown in Figure 1, there are seven carcinogenic acyclic *N*-nitrosamines (NDMA, NMEA, NSAR, NDEA, NDPA, NDELA, and NDBA) to which humans are commonly exposed. We discuss them in order based on their increasing structural complexity. 

### 3.1. N-Nitrosodimethylamine (NDMA)

#### 3.1.1. Exposure and Carcinogenicity

NDMA (**1**, Figure 1) is easily formed by nitrosation with sodium nitrite of an acidified solution of dimethylamine. It was used in industry for several purposes, the most common of which was as a precursor to the rocket fuel 1,1-dimethylhydrazine [20]. However, such uses of NDMA essentially ended after the study of Magee and Barnes [1]. Exposure to NDMA in humans occurs in low concentrations in daily life from food, water, and cosmetics to some contaminated drugs as noted in Section 2. 

The IARC summarized the laboratory animal data of NDMA comprehensively in 1978 and reaffirmed its Group 2A classification in 1987. NDMA is a strong carcinogen primarily targeting the liver but it can also induce kidney tumors when given at high doses [7]. The cancer risk of NDMA exceeds many known strong carcinogens including asbestos, benzo[*a*]pyrene, and polychlorinated biphenyls [9]. A linear dose-response relationship was observed at low dose rates (lower than 1 ppm) in a study with 4080 rats. No indicated safe threshold concentration was observed for NDMA in the rats [21,22]. NDMA is tumorigenic in multiple animal models including Syrian golden hamsters, mastomys, guinea pigs, rabbits, and ducks [7]. The toxicity of NDMA in humans has been demonstrated by some unfortunate poisonings [20]. The currently available data suggest that the IARC classification of NDMA should be re-evaluated. The U.S. government classifies NDMA as “reasonably anticipated to be a human carcinogen” [23]. 

#### 3.1.2. Metabolism

The bioactivation of NDMA to reactive intermediates is primarily catalyzed by P450 2E1 in human liver microsomes [24]. The oxidation of the methyl group (α-methyl hydroxylation) leads to α-hydroxyNDMA (**11**, Figure 1), an unstable and mutagenic intermediate that spontaneously decomposes generating two reactive species—formaldehyde **12** and methyl diazohydroxide **14** [24,25]. Formaldehyde can be oxidized sequentially producing formic acid **13** and CO_2_ [26,27]; methyl diazohydroxide will spontaneously form the highly electrophilic methyldiazonium ion **15** and alkylate DNA [28] or be solvolyzed to methanol [29]. The evolution of ^15^N_2_ from ^15^N-labeled NDMA metabolism suggested that approximately 33–67% of NDMA was metabolized in vitro via α-methyl hydroxylation [30,31,32]. A deuterium isotope effect has been observed with a slight reduction of the NDMA metabolic oxidation rate for NDMA-*d_6_* with the ratio of *V*_H_/*V*_D_ = ~1.2. The effect was more prominent based on the concentrations of DNA adducts formed in the liver and kidneys after oral administration [33]. The methylating species or its immediate precursor **11**, although highly reactive, were sufficiently stable to pass out of rat hepatocytes [34]. The enzymatic denitrosation of NDMA has also been observed in liver microsomes and is considered a detoxification pathway. The major denitrosation product of NDMA is methylamine **17**, formed via the proposed intermediate methyliminium ion **16**; the other product is formaldehyde [35,36,37].

#### 3.1.3. Methyl DNA Adducts Formed by NDMA Metabolism

The mutagenicity and genotoxicity of NDMA are well-established [38]. It alkylates DNA and protein via the two reactive intermediates—methyldiazonium ion **15** and formaldehyde **12**. Methyl DNA adducts formed by the methyldiazonium ion are considered to play a major role in carcinogenesis by NDMA [39]. Formaldehyde, on the other hand, can also form DNA adducts such as cross-links or hydroxymethylene adducts [40].

A comprehensive analysis of methyl DNA adducts in liver DNA was conducted in rats treated with NDMA by a single i.p. dose of 10 mg/kg [41]. Adducts were detected after deglycosylation. Methyl DNA adducts (Figure 3), including *N*3- and *N*7-Me-Ade (**19** and **20**), *O*^2^- and *N*3-Me-Cyt (**21** and **22**), *N*3-, *O*^6^-, *N*7-Me-Gua (**23**, **24** and **27**) and Me-Fapy-Gua (**29**), *O*^2^-, *N*3-, and *O*^4^-Me-Thy (**31**, **33** and **35**), and methyl DNA phosphate adduct **36**, were formed 2 h after administration. The preferential methylation sites in liver DNA were Gua-*N*7, Gua-*O*^6^, and Ade-*N*3. The half-lives of methyl DNA adducts were relatively short, ranging from 4 to 17 h for the methyl base adducts and up to 7 days for the methyl phosphate adduct Tp(Me)T **37** [41]. The adduct profile in NDMA-exposed Syrian golden hamsters was different from that in the rats. *O*^6^-Me-Gua was the most persistent adduct, whereas *N*7-Me-Gua was extensively excised, especially in the liver. Other minor methyl adducts including *N*1-, *N*3-, *N*7-Me-Ade, and *N*3-Me-Gua were also detectable in most hamster tissues [42].

In the rat study, after a 2 mg/kg dose of NDMA by stomach tubing daily (workdays only) for up to 24 weeks, *N*7-Me-Gua **27** predominated in the liver, exceeding that in the kidneys and lungs [43] by approximately 16 times. This is consistent with the liver being the primary tumor site observed in carcinogenicity studies with low doses of NDMA. The other most commonly quantified adduct, *O*^6^-Me-Gua **24**, accounted for 6.7% and 12.0% of *N*7-Me-Gua, 4 h and 24 h after a single s.c. dose of 0.055 mmol/kg NDMA, respectively [44]. Although the levels of *N*7-Me-Gua remained constant in the rat liver, increased excision of *O*^6^-Me-Gua was observed after chronic administration of NDMA [45,46]. This was due to the enhanced activity of *O*^6^-alkylguanine-DNA alkyltransferease (AGT) in the rat liver after repeated low-dose exposure to this carcinogen [47,48,49]. However, a clear interspecies difference has been noted, with a partially depleted AGT activity observed in mice chronically treated with low doses of NDMA [49]. The formation and persistence of *O*^6^-Me-Gua were also greatly affected by the co-treatment with ethanol, a known inhibitor of NDMA hepatic metabolism [50]. A remarkable 10-fold increase was observed in mammary gland DNA adducts in rats co-exposed to NDMA and ethanol. This is probably due to a reduced hepatic first-pass effect, which also resulted in slightly decreased concentrations of the liver DNA adducts [51].

#### 3.1.4. Mutagenicity and Genotoxicity of Methyl DNA Adducts

DNA methylation, especially as related to the activation of proto-oncogenes, has been linked to the induction of pulmonary neoplasia by *N*-nitrosomethylamines [52]. There is convincing evidence that the formation and removal of *O*^6^-Me-Gua **24** (Figure 3) in DNA is related to the risk of tumor induction in different organs [53,54]. *O^6^*-Me-Gua, formed from alkylating agents such as NDMA, is possibly one of the most extensively studied of all DNA adducts. Its persistence in specific rat tissues where tumors developed after treatment with *N*-methyl-*N*-nitrosourea was established in early studies, supporting the hypothesis that it caused miscoding in DNA [55]. Elegant experiments by the Essigmann group demonstrated unequivocally that *O^6^*-Me-Gua causes G to A transition mutations, which is consistent with G-A transition mutations in the *ras* oncogene in laboratory animals treated with NDMA and other methylating carcinogens [56,57]. The repair enzyme AGT can remove the methyl group or other alkyl groups, returning the DNA to its unmodified structure [58].

Methylated thymidines such as *O*^4^-Me-Thd **34** (Figure 3) are poorly repaired [59]. The persistence of the Thd adducts likely contributes to the mutagenicity and carcinogenicity of NDMA. Two adducts, *O*^2^-Me-Thd **30** and *O*^4^-Me-Thd, blocked DNA synthesis and induced A to G transitions mediated by human DNA polymerase κ (pol κ) [60]. This result echoes that of Singer et al. published in 1983 [61].

The methyl Fapy-dGuo adduct **28** blocked eukaryotic high-fidelity polymerases but can be efficiently bypassed by translesion polymerases. Misreplication products, accounting for 8–29% of total extension products, included C to T/G/A mutations and an interesting one-nucleotide deletion [62].

Wang et al. first investigated the mutagenicity of methyl DNA phosphate adduct **36** (Figure 3) [63]. Due to the stereochemistry of the phosphorus atom, two diastereomers are formed upon DNA phosphate methylation. The (*S*)-isomer can be efficiently bypassed, causing TT to GT and GC mutations in the flanking TT dinucleotide site. This mutation was induced in an AGT (also termed as Ada)-dependent manner. On the other hand, the (*R*)-isomer moderately blocked the replication of the DNA synthesis [63].

#### 3.1.5. Methyl DNA Adducts in Human Tissues

The major DNA adducts caused by NDMA incubated with cultured esophagus from human patients were *O*^6^-Me-Gua **24** and *N*7-Me-Gua **27**, in a ratio of 0.3. A 10-fold interindividual variation was observed for adduct levels due to NDMA in human esophageal DNA [64]. In the liver DNA of an NDMA-poisoning victim, *O*^6^-Me-Gua and *N*7-Me-Gua were detected at levels of 273–317 and 1363–1373 μmol/mol Gua, respectively. These adducts were not detected in the liver and kidney DNA from unrelated cases [65].

Methyl DNA adducts have been widely detected in human tissues likely due to exposure to various methylating agents that can originate from multiple sources both endogenously and exogenously [66,67,68]. For example, Foiles et al. developed a monoclonal antibody for *O^6^*-Me-Gua and used it in a competitive enzyme-linked immunosorbent assay coupled with HPLC resulting in the first identification of *O*^6^-Me-Gua in human placental DNA [69]. In Japanese donors, this adduct was detected in both leukocyte and liver DNA. In contrast, *O*^4^-Me-Thy **35** was only detected in the liver DNA [70]. Similarly, in the liver and leukocyte DNA of humans who were not exposed to known alkylating agents, *O*^6^-Me-Gua and *O*^4^-Me-Thy were detected in nearly all the liver DNA samples, at levels of 1.1–6.7 and 0.1–14 adducts/10^7^ nucleotides, respectively. Only *O*^6^-Me-Gua was detected in peripheral leukocyte DNA, accounting for 3.6% of that in the liver [71]. Using a radioimmunoassay with a monoclonal antibody against *O*^6^-Me-dGuo, this adduct has been detected at relatively high levels in the esophagus and liver DNA of esophageal cancer patients in China who were considered exposed to high levels of dietary *N*-nitrosamines [72].

Peter Magee would frequently conclude his lectures by noting that NDMA is a potent hepatocarcinogen in rats, but we have insufficient data to conclude that it is a human carcinogen. Now we have massive biochemical, molecular biological, and laboratory animal testing data in support of its potential human carcinogenicity, resulting in increased regulatory vigilance that has decreased NDMA exposure in most situations to extremely low levels. This appears to be an excellent example of cancer prevention through research and application.

### 3.2. N-Nitrosomethylethylamine (NMEA)

#### 3.2.1. Exposure and Carcinogenicity 

NMEA (**2**, Figure 1) has been found in a variety of processed foods [73,74]. It can be formed as a byproduct during water disinfection with chlorine [75]. It is also one of the contaminants found in some drug products at trace levels [76]. NMEA has been documented as a tobacco constituent at very low concentrations [73,77]. 

The carcinogenicity of NMEA was demonstrated in rats in 1967. Hepatocellular carcinomas are the primary cancer type induced by NMEA administered in drinking water at doses of 1 or 2 mg/kg body weight per day [6]. As part of an extensive program on the chemistry and biology of *N*-nitroso compound [5], Lijinsky et al. showed that deuterated NMEA-*d*_3_ **38** (Figure 4) led to a high incidence of esophageal tumors when compared to undeuterated NMEA administered at an identical dose [78,79].

#### 3.2.2. Metabolism

It is generally considered that NMEA is bioactivated by two α-hydroxylation pathways (Figure 2). When the α-hydroxylation occurs on the ethyl group (as illustrated by **41**), NMEA is converted to methyldiazonium ion **15**, the same intermediate as formed from NDMA metabolism; the other product is acetaldehyde **42**. When the α-hydroxylation occurs on the methyl group (as illustrated by **43**), the ethyldiazonium ion **45** is formed along with formaldehyde **12**. The two diazonium ions **15** and **45** can alkylate DNA and form methyl or ethyl DNA adducts correspondingly. A detailed discussion of the pathway by which α-hydroxylation occurs in the ethyl group can be found in the metabolism study of NDEA in Section 3.4.

#### 3.2.3. DNA Adducts Formed by NMEA Metabolism

After a single i.p. injection of NMEA to F344 rats, *N*7-Me-Gua **27** (Figure 3) was readily detected in the liver, kidneys, and esophageal DNA. However, *O*^6^-Me-Gua **24** was only quantifiable in the liver and kidneys. *N*7-Et-Gua **54** (Figure 5) was also detected in the hepatic DNA [80]. The level of *N*7-Me-Gua in hepatic DNA exceeded *N*7-Et-Gua by 170–200 times. *N*7-Me-Gua occurred at the highest concentration in the liver, followed by the kidneys (15-fold lower), esophagus (100-fold lower), and lung (200-fold lower) [81]. In another study in which F344 rats were administered [1-ethyl-^14^C]NMEA **39** (Figure 4) by a single i.p. injection, *N*7-Et-Gua and *O*^6^-Et-Gua **52** (Figure 5) were the two major ethyl DNA adducts identified in the liver after 4 h exposure. *N*3-Et-Gua **51**, *N*3-Et-Ade **47**, and *N*7-Et-Ade **48** were also identified as minor products resulting from NMEA metabolism [82].

Quantitative differences of methylation and/or ethylation in hepatic DNA were clearly observed in F344 rats administered NMEA or the related *N*-nitrosamines NDMA and NDEA by i.p. injections. Although DNA methylation by NMEA was comparable to that observed upon the administration of an equimolar mixture of NDMA plus NDEA, DNA ethylation by NMEA was ~4-fold lower [83]. Thus, the ratio changes of methyl/ethyl DNA adducts formed by NMEA in the tissue DNA of rats may explain the shift of organ specificity of NMEA-*d*_3_ carcinogenicity reported by Lijinsky et al. [78,79]. After a single i.p. injection of [methyl-^14^C]NMEA-*d*_3_ **40** (Figure 4) to F344 rats, the levels of *N*7-Me-Gua decreased by ~30% in the liver but were 160% greater in the esophagus compared to non-deuterated [methyl-^14^C]NMEA. The increase in esophageal DNA methylation seems to correlate with the increased esophageal carcinogenicity of NMEA-*d*_3_ because of the increased α-hydroxylation of its ethyl group since pharmacokinetic differences between NMEA and NMEA-*d*_3_ have been ruled out [80]. 

### 3.3. N-Nitrososarcosine (NSAR)

#### 3.3.1. Exposure and Carcinogenicity 

NSAR (**3**, Figure 1) has been detected in food such as smoked meat (2–56 μg/kg) [73,84], malt (5.6–11.3 ppb), and beer (trace—6.0 ppb) [85]. It is also one of the carcinogenic components of tobacco smoke (22–460 ng/cigarette) [84] and some smokeless tobacco products (30–550 ng/g) [86,87]. It may also be formed endogenously from nitrosation of sarcosine and dimethylglycine (but not creatine) [88,89,90]. Human exposure to NSAR has been demonstrated by the detection of urinary NSAR, ranging from 0.1 to 3.4 μg/day, in multiple countries [73].

The carcinogenicity of NSAR has been demonstrated in mice and rats. The dietary administration of NSAR caused nasal tumors in mice; oral exposure from drinking water caused esophageal tumors in rats. Liver tumors were observed in newborn mice administered NSAR by i.p. injection [73,84]. Rat studies administering a combination of sarcosine ethyl ester hydrochloride and NaNO_2_ suggested esophageal carcinogenicity of the NSAR derivative *N*-nitrososarcosine ethyl ester **70** (Figure 6) [91,92].

#### 3.3.2. Metabolism

NSAR is not extensively metabolized in rats; 88% of dosed NSAR is excreted unchanged in the urine [93]. Metabolic activation of NSAR starts from α-methyl hydroxylation (Figure 3). The reactive intermediate carboxymethyldiazonium ion **78** and formaldehyde **12** are formed after the spontaneous decomposition of α-hydroxyNSAR **76**. Both products resulting from NSAR α-hydroxylation are able to react with DNA [94]. By analogy to *N*-nitroso-2-oxopropylpropylamine metabolism [95,96,97,98], it also seems possible for NSAR to form the methyldiazonium ion **15** via the intramolecular attack by the diazotate oxygen of **77** on the carbonyl carbon and consecutive decomposition. The proposed methylating agent **15** is likely to be responsible for the observed methyl DNA adducts in DNA exposed to nitrosated glycine derivatives [99,100,101,102,103]. 

#### 3.3.3. Carboxymethylating and Methylating Intermediates Formed by NSAR Metabolism

*O*^6^-Carboxymethylguanine **81** (*O*^6^-CM-Gua, Figure 7) has been detected in DNA treated with mesyloxyacetic acid **71** (Figure 6) [99] and glycine reacted with nitric oxide [104] and related compounds including *N*-(*N*′-acetyl-L-prolyl)-*N*-nitrosoglycine **72** (APNG), azaserine **73** (AS), potassium diazoacetate **74** (KDA), and *N*-nitrosoglycocholic acid **75** [99,100,101,102,103,105]. However, *O^6^*-CM-Gua was not detected in physiological concentrations of glycine and nitric oxide, which does not support the hypothesis that the resulting carboxymethylating species is an etiological agent for human gastrointestinal tumors [104]. In addition, *O*^6^-Me-Gua **24** (Figure 3) has been detected concomitantly with *O*^6^-CM-Gua in vitro; it is proposed to be formed via the putative methyldiazonium ion **15** after decarboxylation (Figure 3) [99,100,101,102,103,104]. This may also partially explain the reported relatively high occurrence of *O*^6^-Me-Gua in human gastrointestinal DNA [72,106,107,108,109,110] since nitrosation of glycine—abundant in food sources—could occur there [111,112].

The other carboxymethyl DNA adducts *N*^6^-(carboxymethyl)-2′-deoxyadenosine **79** (*N*^6^-CM-dAdo, Figure 7), *N*^4^-(carboxymethyl)-2′-deoxycytidine **80** (*N*^4^-CM-dCyd), *N*3-(carboxymethyl)thymidine **83** (*N*3-CM-Thd), and *O*^4^-(carboxymethyl)thymidine **84** (*O*^4^-CM-Thd) have also been detected in vitro in **74** (KDA)-treated calf thymus DNA [113,114]. In cultured human skin fibroblasts and human colorectal carcinoma cells exposed to azaserine **73**, *O*^6^-(carboxymethyl)-2′-deoxyguanosine **82** (*O*^6^-CM-dGuo), *N*^6^-CM-dAdo, and *O*^6^-Me-dGuo were simultaneously measured, with *O*^6^-CM-dGuo predominating among the three determined adducts [115].

#### 3.3.4. Mutagenicity and Genotoxicity of Carboxymethyl DNA Adducts

KDA **74** (Figure 6) caused GC to AT transitions in the p53 gene in equal amounts to GC to TA and AT to TA transversions. This contrasts with the methylating agent methylnitrosourea which causes predominantly GC to AT transitions. The difference is hypothesized to be due to the different mutagenicity of *O*^6^-CM-dGuo **82** compared with *O*^6^-Me-dGuo [105]. 

*O*^6^-CM-dGuo is not a substrate of bacterial and mammalian AGT [103] but can be repaired by human *O*^6^-methylguanine-DNA methyltransferase (MGMT) [116]. It may also be repaired by nucleotide excision repair (NER) [117]. In *E. coli* cells, *O*^6^-CM-dGuo strongly impeded DNA replication. It caused exclusive G to A transitions during the bypass that uniquely requires the involvement of DNA polymerases IV and V [118]. Pol η and ζ were also likely involved in bypassing *O*^6^-CM-dGuo lesions [119].

In a primer extension assay, *N*^4^-CM-dCyd **80** (Figure 7) strongly blocked the extension reaction, causing relatively few C to A or T mutations; *N*^6^-CM-dAdo **79** blocked the extension reaction to a lesser extent but caused a higher frequency of A to T mutations [114]. Pol η was able to readily bypass *N*^6^-CM-dAdo lesions with high fidelity. However, it bypassed *N*^4^-CM-dCyd lesions inefficiently with a substantial frequency of dCMP and dAMP misincorporation [120]. *N*3-CM-Thd **83** and *O*^4^-CM-Thd **84** were highly resistant to the bypass of the yeast Pol η. Preferential T to C mutations caused by those two lesions were observed [120].

The five carboxymethyl DNA adducts *N*^6^-CM-dAdo, *N*^4^-CM-dCyd, *O*^6^-CM-dGuo, *N*3-CM-Thd, and *O*^4^-CM-Thd (Figure 7) were investigated individually for their mutagenicity and genotoxicity in human HEK293T cells. Among the five adducts, *O*^6^-CM-dGuo was moderately genotoxic by blocking DNA replication and weakly mutagenic by inducing 6.4% G to A mutations. In contrast, two Thd adducts, *N*3-CM-Thd and *O*^4^-CM-Thd, were strongly mutagenic, inducing 81% T to A mutations and 68% T to C mutations, respectively [119]. 

#### 3.3.5. Carboxymethyl DNA Adducts in Human Tissues

Based on the evidence of human exposure to NSAR and its potential metabolic activation mechanism, it is reasonable to anticipate the detection of carboxymethyl DNA adducts in humans. In the blood DNA of healthy volunteers restricted to a standardized high-meat diet, *O*^6^-CM-Gua **81** (Figure 7) was detected at levels of 35–80 adducts/10^8^ nucleotides [104]. The formation of *O*^6^-CM-Gua in the colonic exfoliated cells of healthy volunteers has been positively associated with the consumption of red meat. Levels of *O*^6^-CM-Gua were also positively correlated with the concentrations of apparent total fecal *N*-nitroso compounds [121]. These results provide some evidence supporting the etiological role of dietary *N*-nitroso compounds such as NSAR or nitrosated glycine derivatives, which can decompose to form carboxymethylating and methylating agents, in human gastrointestinal carcinogenesis. However, further studies are required. 

Interestingly, we could not detect *N*7-(carboxymethyl)guanine **85** (*N*7-CM-Gua, Figure 8) in any human liver samples but its analog *N*7-(2′-carboxyethyl)guanine **86** (*N*7-CE-Gua) occurred at a relatively high mean level of 373 ± 320 fmol/μmol Gua in 100% of the same human livers. One possible source of *N*7-CE-Gua is 3-(methylnitrosamino)propionic acid **87** (MNPA) metabolism [94].

### 3.4. N-Nitrosodiethylamine (NDEA)

#### 3.4.1. Exposure and Carcinogenicity 

Human exposure to NDEA (**4**, Figure 1) occurs through water, food, and cosmetics [18,73]. Some batches of drugs including irbesartan, losartan, and valsartan have been recalled due to contamination with NDEA since 2018 [16]. It has also been identified in tobacco smoke, with greater concentrations found in sidestream smoke compared to mainstream smoke [73]. However, the levels of NDEA in current cigarette smoke and smokeless tobacco products are extremely low [77].

The carcinogenicity of NDEA has been demonstrated in multiple laboratory animal species including mice, rats, Syrian golden hamsters, guinea pigs, rabbits, dogs, pigs, and monkeys [73]. Tumors caused by NDEA primarily occurred in the liver, often with lung metastases in mice and rats after oral administration [73]. Peto et al. conducted a large study of rats treated with NDEA in drinking water for their lifetime starting from 6 weeks of age. Various types of liver cancers were observed primarily in the NDEA-treated rats; esophageal cancers were also observed exclusively with NDEA but not NDMA treatment. A few nasopharyngeal tumors were also caused by NDEA in the rats [21,22]. The IARC upgraded NDEA to Group 2A in 1987, but as is the case with NDMA, re-evaluation may be appropriate.

#### 3.4.2. Metabolism

NDEA metabolic activation for carcinogenicity is principally catalyzed by P450 2E1 and P450 2A6 [24,122,123]. When the P450-catalyzed hydroxylation occurs at the α-carbon of the ethyl group of NDEA, the electrophilic ethyldiazonium ion **45** (Figure 4) is formed after decomposition of the unstable intermediate ethyl diazohydroxide **44**. Intermediate **44** reacts with DNA producing ethyl DNA adducts such as *N*7-Et-Gua and *O*^6^-Et-Gua. Acetaldehyde **42** is also formed by NDEA α-hydroxylation [124]. When the β-carbon is hydroxylated, the reactive intermediate 2-hydroxyethyldiazonium ion **100** can be formed by a secondary α-hydroxylation on the other ethyl group of NDEA (as illustrated by **90**). The 2-hydroxyethyldiazonium ion **100** alkylates DNA and forms DNA adducts such as *N*7-HOEt-Gua **69** (Figure 5) that have been detected in the hepatic DNA of NDEA-treated rats [82]. The denitrosation reaction competes with bioactivation in NDEA metabolism, suggesting a dual role played by P450s in the biotransformation of NDEA [35].

#### 3.4.3. Ethyl DNA Adducts Formed by NDEA Metabolism

Ethyl DNA adducts are readily formed by NDEA metabolism in vivo after a single administration of the carcinogen [125,126]. Although the main ethyl DNA adducts (as shown in Figure 5) were ethyl DNA phosphate adducts B_1_p(Et)B_2_ **61** (58%), ethyl DNA base adducts after deglycosylation were also produced as expected. *N*7-Et-Gua **54** (12%), *O*^6^-Et-Gua **52** (8%), *O*^2^-Et-Thy **55** (7%), and *N*3-Et-Ade **47** (4%) represented the most abundant ethyl DNA base adducts in the same sample mixture. The other minor base adducts observed were *N*1-Et-Ade **46**, *N*7-Et-Ade **48**, *O*^2^-Et-Cyt **49**, *N*3-Et-Cyt **50**, *N*3-Et-Gua **51**, *N*3-Et-Thy **57**, and *O*^4^-Et-Thy **59** [125,126]. In addition, trace levels of *N*7-HOEt-Gua **69** were also detected in the hepatic DNA of rats exposed to NDEA, comprising less than 2% of *N*7-Et-Gua **54** [82]. 

Variation in ethyl DNA base adduct accumulation has been investigated in the liver DNA of rats exposed to NDEA in drinking water for 77 days [127,128]. *O*^4^-Et-Thd **60** accumulated in the first 28 days and persisted at such levels until the end of the experiment. *O*^6^-Et-dGuo **53**, in contrast, did not accumulate after repeated exposure to NDEA. The concentration of *O*^6^-Et-dGuo was highest after 2 days of administration of NDEA and decreased throughout the 77-day study course [127,128]. This was attributed to a rapid repair mechanism for *O*^6^-Et-dGuo when compared with *O*^4^-Et-Thd [129].

#### 3.4.4. Mutagenicity and Genotoxicity of Ethyl DNA Adducts

The dynamic change in the accumulation of *O*^6^-Et-dGuo and *O*^4^-Et-Thd after continued exposure to NDEA clearly indicates the different repair mechanisms of these ethyl DNA adducts. *O*^6^-Et-dGuo can be repaired by AGT without the excision of the DNA base [130]. On the contrary, *O*^4^-Et-Thd is not a good substrate of rat liver AGT; the half-life of *O*^4^-Et-Thd reaches 11–19 days in vivo [41,129,131,132]. A similar result was also observed for *O*^2^-Et-Thd **56** (Figure 5) [133]. The DNA repair protein AGT repairs *O*^4^-Et-Thd, however, to a lesser extent compared to its analog *O*^4^-Me-Thd **34** (Figure 3) [134,135,136].

Diverse mutagenic consequences have been observed for the 3 regioisomers of ethyl thymidine adducts. *O*^4^-Et-Thd, a major-groove lesion, induces a large number of A to G transitions when incorporated into polynucleotides. However, *O*^2^-Et-Thd (a minor-groove lesion) was only slightly mutagenic and *N*3-Et-Thd **58** (Figure 5) did not induce mutations [137]. *O*^4^-Et-Thd is also genotoxic by moderately blocking DNA replication with a bypass efficiency of 20–33% in human cells [138].

The DNA polymerases responsible for bypassing the three regioisomeric ethyl thymidine lesions have been extensively investigated. DNA polymerase I and RNA polymerase II can recognize the ethyl Thd lesions and direct the dGMP misincorporation opposite to *O*^4^-Et-Thd but not *O*^2^-Et-Thd [139,140]. Human Pol η similarly can bypass all 3 ethyl Thd lesions and generate the replication product with a substantial frequency of A to G transitions caused by *O*^4^-Et-Thd [141,142,143]. In *E. coli* cells, both Pol IV and Pol V are essential for the misincorporation of dCMP opposite to *O*^2^-Et-Thd, whereas Pol V is necessary for the T to A transversions caused by this lesion [144]. In human cells, Pol η and ζ are the predominant polymerases mainly responsible for bypassing *O*^2^- and *O*^4^-Et-Thd, and causing the respective T to A/G and T to C mutations [138,145].

#### 3.4.5. Ethyl DNA Adducts in Human Tissues

Ethyl DNA adducts have been detected in various human tissues. They may arise from the metabolism of the *N*-nitrosamines NDEA and NMEA, or from structurally unknown directly acting ethylating agents present in tobacco and tobacco smoke [146]. Levels of *O*^2^-, *N*3-, and *O*^4^-Et-Thd (**56**, **58,** and **60**, Figure 5) in the leukocyte DNA of 20 smokers were 44.8 ± 52.0, 41.1 ± 43.8, and 48.3 ± 53.9 adducts/10^8^ nucleotides, respectively, significantly exceeding those in the 20 nonsmokers. The formation of each ethyl thymidine adduct was statistically associated with that of the other two ethyl Thd adducts [147]. *O*^4^-Et-Thd was detected in the lower respiratory tract DNA of smokers but not in nonsmokers [148]. *O*^4^-Et-Thd levels in the lung DNA of smokers (3.8 adducts/10^8^ nucleotides) were higher (*p* < 0.01) than in nonsmokers (1.6 adducts/10^8^ nucleotides) [149]. The smoking-related formation of this adduct in lung DNA was further confirmed in Hungarian lung cancer patients [150]. 

Similarly, levels of *N*3-Et-Ade **47** and *N*7-Et-Gua **54** (Figure 5) were also significantly higher in the leukocyte DNA from smokers (16.0 ± 7.8 and 9.7 ± 8.3 adducts/10^8^ nucleotides, respectively) than those from nonsmokers (5.4 ± 2.6 and 0.3 ± 0.8 adducts/10^8^ nucleotides, respectively). The levels of *N*3-Et-Ade and *N*7-Et-Gua were positively correlated [151]. *N*3-Et-Ade and *N*7-Et-Gua were also detected in the salivary DNA of smokers and nonsmokers. The occurrence of *N*7-Et-Gua in the saliva of smokers (14.1 ± 8.2 adducts/10^8^ nucleotides) was significantly higher than that of nonsmokers (3.8 ± 2.8 adducts/10^8^ nucleotides, *p* < 0.0001). The levels of *N*7-Et-Gua were also strongly associated with tobacco smoking [152]. However, there are some contradictory results that question the robustness of applying ethyl DNA adducts as biomarkers for smoking-related cancer etiology studies. We have quantified the levels of *N*7-Et-Gua in human leukocyte DNA from 30 smokers and 30 nonsmokers. No statistical difference was observed in the levels of this adduct in the leukocyte DNA from the two subject groups (smokers: 49.6 ± 43.3 fmol/μmol Gua; nonsmokers: 41.3 ± 34.9 fmol/μmol Gua) [153]. 

In addition to human tissue DNA, ethyl DNA adducts have been detected in human urine showing a potential correlation with smoking status. Urinary excretion of *N*3-Et-Ade has been observed to increase 5–8-fold with tobacco smoking [154,155] but not from dietary origin [156]. At 50%, the mean level of *N*-terminal *N*-ethylvaline in the hemoglobin of smokers was also significantly higher than in the nonsmokers [157].

Other than smoking, dietary exposure to ethylating agents derived from *N*-nitrosamines or possibly other sources may play an important role in human carcinogenesis, especially in some particular geographic areas. In Linxian, the incidence of esophageal cancer was comparably higher than its surrounding area in the same region of China. In the esophageal and hepatic DNA of esophageal cancer patients from Linxian, relatively high levels of *O*^6^-Et-dGuo **53** (Figure 5) were detected by radioimmunoassay. This appeared to be consistent with the relatively high exposure levels of dietary *N*-nitrosamines in those who lived in this area [72].

The endogenous formation of ethylating agents may be important in the formation of ethyl DNA adducts in the human liver. In the liver DNA of 15 autopsy specimens, *O*^4^-Et-Thd occurred at levels of 0.5–140 adducts/10^7^ nucleotides [71]. It was not detected in the peripheral leukocyte DNA in the same study [70,71]. *N*7-Et-Gua was detected in 25 of 26 human hepatic DNA samples, occurring at a level of 42.2 ± 43.0 fmol/μmol Gua [158]. 

### 3.5. N-Nitroso-di-n-propylamine (NDPA)

#### 3.5.1. Exposure and Carcinogenicity

Human exposure to NDPA (**5**, Figure 1) may arise from the consumption of drinking water, food, and beverages, contact with pesticides and wastewater, or from endogenous formation due to the use of nitrite- or secondary amine-containing food or drugs [73,159]. 

The carcinogenicity of NDPA has been demonstrated in laboratory animal studies with rats, mice, hamsters, and monkeys. After the oral administration of NDPA to rats, tumors primarily occurred in the liver, nasal cavity, and esophagus [73,159]. 

#### 3.5.2. Metabolism

NDPA is metabolized via α-, β-, and γ-hydroxylation of the propyl group (Figure 5). Of the three metabolic pathways, α-hydroxylation is regarded as the primary route for NDPA bioactivation [159].

Catalyzed primarily by P450 2E1 and P450 2B1 [160,161], *N*-nitroso-1-hydroxypropylpropylamine **101** is formed by the α-hydroxylation of NDPA. It decomposes to generate the reactive intermediates propyl diazohydroxide **102** and propionaldehyde **106** [162]. The diazohydroxide further reacts forming the electrophilic carbocations **104** and **107** via the intermediate propyldiazonium ion **103**. The solvolysis products of the two carbocations—1-propanol **105** and 2-propanol **108**—have been detected in vitro [163,164].

The β-hydroxylation of NDPA generates *N*-nitroso-2-hydroxypropylpropylamine **109** (NHPPA); its glucuronide is excreted, accounting for 5% of the administered NDPA in 24 h rat urine [165]. NHPPA can be further oxidized to *N*-nitroso-2-oxopropylpropylamine **110** (NOPPA) [163,165]. NOPPA can be reduced back to NHPPA, accounting for nearly 50% of the total dose of NOPPA in 24 h rat urine [165,166]; carbonyl reduction was similarly observed in the metabolism of relevant metabolites such as *N*-nitroso-(2-hydroxypropyl)-(2-oxopropyl)amine **118** (Figure 9) [167]. NOPPA also undergoes a secondary α-hydroxylation (as illustrated by **111**), generating the methyldiazonium ion **15** and acetic acid **113** via an oxadiazoline intermediate resulting from a spontaneous intramolecular attack by the diazotate oxygen on the carbonyl carbon of **112** [95,96,97,98]. P450 2E1 and P450 2B1 play a major role in the consecutive hydroxylation of NOPPA [97].

The γ-hydroxylation of NDPA forms *N*-nitroso-3-hydroxypropylpropylamine **114** and *N*-nitrosopropyl-(carboxyethyl)amine **115**. They were detected as minor metabolites of NDPA in isolated rat hepatocytes [168].

Considering the structural similarity of NDPA and other *N*-alkylnitrosamines such as NDMA and NDEA, P450-catalyzed denitrosation is likely to occur through a radical mechanism ultimately forming propylamine, propionaldehyde, and nitrate [37,169].

#### 3.5.3. DNA Adducts Formed by NDPA Metabolism

In 1971 and 1973, Kruger first investigated the alkylation of nucleic acids using [^14^C]NDPA. The simultaneous detection of *N*7-Me-Gua **27** (Figure 3) and *N*7-(*n*-propyl)guanine **119** (*N*7-*n*-Pr-Gua, Figure 10) in the liver RNA of rats treated with NDPA strongly suggested the metabolism of NDPA by both α- and β-hydroxylation pathways [96]. [^14^C]*N*7-Me-Gua was only detectable in rat liver RNA and DNA when the rats were treated with [α-^14^C]NDPA, in agreement with the methylating agent arising from the β-hydroxylation of NDPA [170]. When the rats were treated with [β-^14^C]NDPA, only [^14^C]*N*7-*n*-Pr-Gua was detectable, consistent with the metabolism mechanism of NDPA α-hydroxylation [96]. Similarly, [^3^H]*N*7-Me-Gua was detected in the rat liver after application with [^3^H]NOPPA [171].

Alkylation of DNA has been clearly demonstrated using [α-^14^C]NDPA in vitro [172]. However, the chemical characterization of NDPA-DNA adducts is limited. Only some related studies have been reported. Kokkinakis in 1992 reported a study of methyl and hydroxypropyl DNA adducts in the tissues of hamsters and rats after a single s.c. dose of ^3^H-labeled *N*-nitroso-bis(2-hydroxypropyl)amine **116** (Figure 9). Methyl DNA adducts were preferentially formed over hydroxypropyl DNA adducts at low doses (100–500 mg/kg body weight) but became secondary at higher doses. Both adducts occurred at their highest concentrations in the liver, the primary metabolic activation site. The methyl DNA adducts identified in this study were *N*7-Me-Gua **27** and *O*^6^-Me-Gua **24** (Figure 3); the hydroxypropyl DNA adducts were *N*7-(2-hydroxypropyl)guanine **120**, *O*^6^-(2-hydroxypropyl)guanine **121,** and *O*^6^-(1-methyl-2-hydroxyethyl)guanine **122** (Figure 10) [173]. The formation of *N*7-Me-Gua and *O*^6^-Me-Gua was also confirmed in hamsters and rats treated with the NDPA derivatives *N*-nitroso-bis(2-oxopropyl)amine **117** and *N*-nitroso-(2-hydroxypropyl)-(2-oxopropyl)amine **118** (Figure 9), both of which are pancreatic carcinogens in hamsters [174,175]. Adduct **120** was also detected in the tissues of hamsters and rats treated with **117** and **118** [175]. 

### 3.6. N-Nitrosodiethanolamine (NDELA)

#### 3.6.1. Exposure and Carcinogenicity 

NDELA (**6**, Figure 1) is an environmentally prevalent *N*-nitrosamine found in cosmetics. It can also be detected in some food products, synthetic cutting fluids, and tobacco and tobacco smoke [176,177,178,179].

The carcinogenicity of NDELA has been extensively studied in rats, mice, and hamsters [73,179]. After oral administration, NDELA induced primarily liver tumors in rats. It also induced lung tumors in mice and some nasal tumors in rats. In hamsters, NDELA treatment resulted in nasal cavity tumors and tracheal tumors regardless of the administration pathways (s.c. injection, topical application, and oral swabbing) [73,179].

#### 3.6.2. Metabolism

The distribution of NDELA in Osborne-Mendel rats has been studied using two administration pathways. After oral administration, NDELA was absorbed and distributed rapidly and reached a peak concentration at 8 h; after topical application, NDELA was slowly absorbed but rapidly distributed as when dosed orally. NDELA was excreted mainly in the urine as the unchanged form and one metabolite [180,181]. The excreted NDELA also represented a high percentage (60–90%) in the urine of male Sprague-Dawley rats administered NDELA in drinking water [182]. A similar high urinary excretion rate (73–89%) of unchanged NDELA was also determined in rats treated percutaneously and intratracheally [19]. However, the absorption rate of NDELA was significantly lower in Syrian golden hamsters. After s.c. injection, 49% and 11% of the dose appeared in the urine and feces, respectively, in 16 h; 34% and 6% after oral swabbing; and only 21% and 4% were detected after skin application [183].

An investigation of the rat urinary metabolites of NDELA suggested only one compound containing the nitroso moiety, which was identified as *N*-nitroso-(2-hydroxyethyl)glycine **135** (NHEG, Figure 6). It represented 6% of the dosed NDELA in the rats [184]. The glucuronide of NDELA was also identified in rat urine after gavage [185]. However, no sulfate derivatives were observed even though NDELA sulfate was considered a possible activated metabolite that could react with DNA to form 2-hydroxyethyl adducts (as shown in Figure 5) [185,186,187]. *N*-Nitroso-2-hydroxymorpholine **137** (NHMOR) was also observed as a minor metabolite in vitro in rat liver S9 supernatant [188,189].

The formation of NHEG **135** and NHMOR **137** strongly implies the important role that β-hydroxylation may play in the bioactivation of NDELA in vivo (Figure 6). This reaction is catalyzed primarily by P450 2E1 [188]. NHMOR can be further metabolized by α-hydroxylation on the two methylene groups. When the α-hydroxylation occurs on the 3-carbon, the major metabolite has been identified as glyoxal **133** formed via **139** and **140**; when the α-hydroxylation occurs on the 5-carbon, the major metabolite is 2-acetoxyacetaldehyde **144** formed via **142** and **143** [188]. It is noteworthy that NHMOR also arises from NMOR metabolism which is discussed in detail in Section 4.3.2.

However, several lines of evidence raise questions regarding the importance of β-hydroxylation in the carcinogenesis of NDELA. Although it is a stable precursor to potential DNA alkylating agents such as glyoxal, NHMOR was inactive or marginally carcinogenic to rats or mice when administered in drinking water [190]. Only the glyoxal-deoxyguanosine adduct **145** (*N*1,*N*^2^-glyoxal-dGuo, Figure 11) was observed in vitro in NHMOR-incubated calf thymus DNA (up to 48 h) and in vivo in the liver DNA of rats given NHMOR at a single dose by gavage for 4 h [191]. The levels of *N*1,*N*^2^-glyoxal-dGuo formed by NHMOR were also lower than those formed by NDELA [192]. Those findings taken together suggest that additional metabolic activation pathways such as α-hydroxylation are involved in the carcinogenesis of NDELA [193].

Catalyzed primarily by P450 2E1, hydroxylation occurs on the α-carbon of NDELA and forms the 2-hydroxyethyldiazonium ion **127** and glycolaldehyde **130** after spontaneous decomposition [188,194]. The 2-hydroxyethyldiazonium ion forms the carbocation **128** after the loss of H_2_O and subsequently yields the solvolysis product ethylene glycol **129**, which can also undergo microsome-mediated oxidation to glyoxal **133**. The carbocation **128** also undergoes elimination and forms acetaldehyde **42** [188,195]. *O*^6^-(2-Hydroxyethyl)-2′-deoxyguanosine **68** (*O*^6^-HOEt-dGuo, Figure 5) arising from NDELA metabolism has been detected in rat liver DNA; it is not derived from NHMOR metabolism [191]. Deuteration on the α-carbon of NDELA greatly decreased the formation of glycolaldehyde **130** and *O*^6^-HOEt-dGuo **68**, whereas β-deuteration oppositely affected the formation of those products. This further suggests that *O*^6^-HOEt-dGuo mainly results from NDELA α-hydroxylation [191,194]. One unexpected finding was that glycolaldehyde **130**, other than being classically converted to glycolic acid **131** and oxalic acid **132**, was also transformed to glyoxal **133** via the catalysis of P450 2E1 [188,194]. The collective evidence indicates the necessity of the α-hydroxylation of NDELA for DNA adduct formation. 

#### 3.6.3. DNA Adducts Formed by NDELA Metabolism

*O*^6^-HOEt-dGuo **68** and *N*7-HOEt-Gua **69** (Figure 5) were characterized in the reaction mixture of *N*-nitroso-3-acetoxy-2-hydroxymorpholine **138** (Figure 6) with dGuo. However, neither of these adducts was detected in the reaction mixture of *N*-nitroso-5-acetoxy-2-hydroxymorpholine **141** with dGuo. This suggested that the α-hydroxylation of NHMOR on the 5-carbon was unlikely to produce DNA adducts, whereas NHMOR α-hydroxylation on the 3-carbon might yield a carcinogenic outcome [195]. However, this seems not to be supported by an in vivo study, in which no 2-hydroxyethyl guanine adducts were detected in the liver DNA of rats treated with NHMOR [188]. In contrast, 2-hydroxyethyl guanine adducts were tentatively identified in the hydrolysates of the liver DNA of rats treated with NDELA by gavage [185]. One adduct was later characterized as *O*^6^-HOEt-dGuo using the synthesized authentic standard. It was detected in vivo in the liver DNA of rats treated with NDELA [191,194,196]. Taken together, 2-hydroxyethyl adducts formed by NDELA are reasonably considered to result from the α-hydroxylation pathway rather than the β-hydroxylation pathway.

Glyoxal DNA adducts formed by NDELA and its analogs were readily detected in vitro and in vivo. In the reaction with dGuo in vitro, *N*1,*N*^2^-glyoxal-dGuo **145** (Figure 11) was formed as the major adduct (65%) by *N*-nitroso-3-acetoxy-2-hydroxymorpholine **138**; *N*1,*N*^2^-etheno-dGuo **146** was formed as the major adduct (44%) by the less-reactive *N*-nitroso-5-acetoxy-2-hydroxymorpholine **141** [195]. *N*1,*N*^2^-glyoxal-dGuo was also detected in the liver DNA of rats treated with NDELA or NHMOR [192,196]. A few analogs of NDELA also caused the same *N*1,*N*^2^-glyoxal-dGuo adduct in the rat liver DNA [192]. However, due to the complex potential origins of glyoxal, the formation of *N*1,*N*^2^-glyoxal-dGuo does not necessarily reflect the preference of metabolic pathways of NDELA.

### 3.7. N-Nitrosodi-n-butylamine (NDBA)

#### 3.7.1. Exposure and Carcinogenicity 

NDBA (**7**, Figure 1) has been found in agricultural products, fish, processed meats, seasonings, and contaminated water [12,73]. It was also formed at trace levels during the production of the drug ranitidine [197]. Some early data suggested the presence of NDBA in tobacco smoke but with no clear evidence in recent studies [73].

NDBA was classified by IARC in 1978 and reaffirmed in 1987 as a group 2B carcinogen. It is carcinogenic to the esophagus and bladder of laboratory animals including mice, rats, Syrian golden hamsters, and guinea pigs. It also causes liver and forestomach tumors [73,198,199,200,201,202]. In 1983, Lijinsky and Reuber reported that NDBA, even though much weaker than NDPA, induced liver tumors in 60% of rats administered by gavage for 83 weeks. It also induced forestomach (50%) and bladder tumors (35%) [203]. Even after a short-term (2 weeks) exposure to NDBA, preneoplastic lesions were positively found in the liver, esophagus, forestomach, and bladder of rats 52 weeks post-treatment [204]. Mice with a p53 gene knockout had increased susceptibility to esophageal and bladder carcinogenesis caused by NDBA [205].

#### 3.7.2. Metabolism

By analogy to other *N*-nitrosodialkylamines, NDBA requires metabolic activation to exert its carcinogenicity. Four metabolic pathways can occur in NDBA metabolism (Figure 7). They are α-, β-, γ-, and δ-hydroxylation of the butyl group of NDBA, among which α-hydroxylation has been suggested to be primarily involved in NDBA carcinogenesis [206].

In an incubation mixture of NDBA with rat liver microsomes, the principal metabolite retaining the *N*-nitroso group was the γ-hydroxylation product *N*-nitroso-(3-hydroxybutyl)butylamine **157**. The α-hydroxylation products butyraldehyde **152** and/or 1-butanol **154** and 2-butanol **155** were also detected, indicating the formation of the butyl carbocations **150** and **151** [206,207]. In rats, NDBA was extensively metabolized with no unchanged compounds being detected in the urine. Two major metabolites *N*-nitroso-(3-carboxypropyl)butylamine **161** and *N*-nitroso-(3-hydroxybutyl)butylamine **157** resulted from δ- and γ-hydroxylation, respectively. The third minor metabolite was the β-hydroxylation product *N*-nitroso-(2-hydroxybutyl)butylamine **156**. All metabolites were detected in the urine as such and as their gluconides [208]. One of the major urinary metabolites **161** arises from the consecutive oxidation of the initial δ-hydroxylation product *N*-nitroso-(4-hydroxybutyl)butylamine **160** (BBN), a urothelial carcinogen [209]. It can be further converted to several minor metabolites such as **162**, **163,** and **164** by β-oxidation and subsequent biotransformations [210,211]. In contrast to rats, the primary metabolite of NDBA in hamsters was the glucuronide of *N*-nitroso-(3-hydroxybutyl)butylamine **157** [212].

In the urine of rats treated with NDBA, the end products of NDBA-GSH conjugates were also detected. They are *N*-acetyl-*S*-butyl-L-cysteine **153** resulting from α-hydroxylation of NDBA and *N*-acetyl-*S*-3-hydroxybutyl-L-cysteine **158** and *N*-acetyl-*S*-3-oxobutyl-L-cysteine **159** that are hypothesized to result from γ-hydroxylation of NDBA followed by a secondary α-hydroxylation [213].

#### 3.7.3. DNA Adducts Formed by NDBA Metabolism

The butyl carbocation **150** (Figure 7) resulting from the α-hydroxylation of NDBA is considered to be the alkylating agent which attacks DNA forming *n*-butyl DNA adducts. In the liver DNA of rats treated with 185 mg/kg NDBA by a single i.p. dose, *O*^6^-(*n*-butyl)guanine **123** (*O*^6^-*n*-Bu-Gua, Figure 10) has been detected at a concentration of 0.34 μmol/mol Gua [214]. 

The sequential α-hydroxylation of NDBA metabolites such as BBN **160** can form the other alkylating species and react with DNA to cause lesions. In the urothelial and hepatic DNA of rats, both *O*^6^-*n*-Bu-Gua **123** and *O*^6^-(4-hydroxybutyl)guanine **124** (*O*^6^-(4-OH-*n*-Bu)-Gua, Figure 10) were detected at 17.9 ± 7.23 and 12.2 ± 7.01 μmol/mol Gua, respectively, after 24 h treatment with a single oral dose of 120 mg BBN. *O*^6^-*n*-Bu-Gua did not accumulate after repeated exposure to a lower dose of BBN; *O*^6^-(4-OH-*n*-Bu)-Gua was not detected in the same study [215].

## 4. Metabolic Activation and DNA Interactions of Carcinogenic Cyclic *N*-Nitrosamines

Three carcinogenic cyclic *N*-nitrosamines (NPYR, NPIP, and NMOR, Figure 1) are discussed here due to their common human exposure. They are introduced in order based on their increasing structural complexity. 

### 4.1. N-Nitrosopyrrolidine (NPYR)

#### 4.1.1. Exposure and Carcinogenicity

NPYR (**8**, Figure 1) is a simple symmetric cyclic *N*-nitrosamine. It has been extensively investigated as a model compound for studies of *N*′-nitrosonornicotine (NNN), and in its own right due to its common occurrence in food (1.5 ± 0.2 ng/g) and water (5.5 ± 2.6 ng/L) [9]. It is also present in tobacco smoke and smokeless tobacco products [73,77,216]. Endogenous nitrosation is considered to be another possible pathway to form NPYR, due to the relatively high human exposure to its precursor pyrrolidine, which is excreted to the extent of ~20 mg per day in the urine [217].

NPYR is a strong hepatic carcinogen in mice and rats. The IARC reviewed some early data on NPYR carcinogenicity in 1978 [73]. Since then, some new data have been reported. In rats administered NPYR in drinking water, it primarily caused hepatocellular carcinomas, many of which metastasized [218,219]. A dose-response relationship of hepatic tumor formation was observed in rats administered NPYR in drinking water [220]. NPYR also induced lung tumors in A/J mice after i.p. injection [221]. In Syrian golden hamsters, NPYR induced tracheal and nasal cavity tumors when administered intraperitoneally [222] but tracheal papillomas and hepatic neoplastic nodules when fed in the diet [223]. 

#### 4.1.2. Metabolism

NPYR is metabolized extensively in rats, with only ~1% of unchanged NPYR detected in the urine [224]. It is ultimately excreted as volatiles such as CO_2_ and N_2_ [224,225]. To exert its carcinogenicity, NPYR requires metabolic activation primarily catalyzed by P450 2E1 [224,226]. The identification of 2-hydroxytetrahydrofuran **171** (2-hydroxyTHF, Figure 8) [227,228,229], crotonaldehyde **177** [230], and 3-hydroxy-1-nitrosopyrrolidine **178** (3-hydroxyNPYR) [231,232] as metabolites of NPYR in rats indicated the occurrence of the α- and β-hydroxylation pathways of NPYR metabolism. The nearly diminished carcinogenicity of the NPYR analog 2,5-dimethyl-*N*-nitrosopyrrolidine in rats suggested the importance of α-hydroxylation for the metabolic activation of NPYR to express its ultimate carcinogenicity. 

Catalyzed primarily by P450 2E1 [224,226], NPYR is metabolized via α-hydroxylation to form 2-hydroxyNPYR **167**, which decomposes spontaneously to the 4-oxobutyldiazonium ion **169** through the intermediate 4-oxobutyl diazohydroxide **168**. After the loss of one molecule of N_2_ from the diazonium ion, carbocations **172** and **174** and the oxonium ion **170** are suggested to be the reactive intermediates in forming the solvolysis products 4-hydroxybutanal **173** in equilibrium with 2-hydroxyTHF **171** and 3-hydroxybutanal **175** [227,228,229] and the elimination product crotonaldehyde **177** [230,233,234,235]. Among those metabolites, 2-hydroxyTHF **171** is the most prevalent one [227].

α-Hydroxylation of NPYR predominated in explanted esophagus from rats and humans [236], cultured human colon [237], and cultured bladder cells from rats and humans [238]. The rate of NPYR α-hydroxylation by rat hepatic microsomes was 1.43 nmol/min/mg protein [228]; this rate was lower in human hepatic microsomes (0.68 nmol/min/mg protein) [239]. The apparent *K*_m_ for rat lung microsomes and post-microsomal supernatant was approximately 20 mM but was much smaller for rat liver microsomes (0.36 mM) [229,240]. NPYR α-hydroxylation was inducible in rats and hamsters that were pre-treated with Aroclor or ethanol [228,241,242]. 

#### 4.1.3. DNA Adducts Formed by NPYR Metabolism

As depicted in Figure 8, the butanal carbocation **172** is considered to form the oxonium ion **170** and the isomeric carbocation **174** after rearrangement. All 3 reactive species can react with DNA and form the corresponding DNA adducts. Crotonaldehyde **177**, the elimination product of the isomeric carbocation **174**, also reacts with DNA. 

##### DNA Adducts Formed by NPYR-Derived Carbocations

In 1982, Hunt and Shank first reported the detection of a fluorescent adduct, which was formed dose-dependently in the liver DNA of rats treated with NPYR [243]. We later structurally characterized this adduct to be 2-amino-6,7,8,9-tetrahydro-9-hydroxypyrido [2,1-*f*]purine-4(3*H*)-one **179** (*N*7,8-butano-Gua, Figure 12) [244], which was likely formed by simultaneous alkylation at Gua-*N*7 and cyclization at Gua-C8 [245,246]. *N*7,8-Butano-Gua was the predominant adduct formed among the NPYR-derived DNA adducts in the liver DNA of rats (see Table 1) [247]. It occurred most abundantly in the target organ liver but was also detected in the kidneys and lungs of NPYR-treated mice, rats, and hamsters [248]. In rats treated with NPYR by intragastric administration, this adduct peaked at 12–24 h after dosing in a dose-dependent manner and persisted for at least 3 days [248]. *N*7,8-Butano-Gua occurred to a higher extent in the RNA than in the DNA of rat liver (see Table 1), suggesting that RNA could be superior for biomarker studies of this lesion [249]. It was also detected in rat urine [249].

Two acyclic adducts—*N*7-(4-oxobutyl)guanine **180** and *N*7-(3-carboxypropyl)guanine **181**—can also be formed by the carbocation resulting from NPYR α-hydroxylation. They were both identified in vitro in calf thymus DNA incubated with α-acetoxy-*N*-nitrosopyrrolidine **165** (α-acetoxyNPYR, Figure 8); *N*7-(4-oxobutyl)guanine predominated among these two adducts [245,246]. There was little evidence in support of the cyclization of *N*7-(4-oxobutyl)guanine to *N*7,8-butano-Gua **179** [245]. *N*7-(4-Oxobutyl)guanine occurred at about one-third of the level of *N*7,8-butano-Gua, at a concentration of 643 ± 9 μmol/mol Gua in the hepatic DNA of rats administered NPYR by i.p. injection [250]. A similar concentration (603 μmol/mol Gua) was also observed in the liver DNA but not the liver RNA of rats administered NPYR by gavage [249].

##### DNA Adducts Formed by an NPYR-Derived Oxonium Ion

The NPYR-derived oxonium ion **170** reacts with DNA forming a group of tetrahydrofuranyl DNA adducts, some of which have been identified both in vitro and in vivo. Quantitation of those adducts has been performed by reducing them to the corresponding 4-hydroxybutyl adducts (see Table 1).

In the reaction of dGuo and calf thymus DNA with α-acetoxyNPYR **165**, *N*^2^-(tetrahydrofuran-2-yl)-2′-deoxyguanosine **185** (*N*^2^-THF-dGuo, Figure 12) was characterized as the first identified tetrahydrofuranyl DNA adduct derived from NPYR metabolism [253,254]. The levels of this adduct exceeded those of other adducts formed by α-acetoxyNPYR in the in vitro DNA hydrolysate samples [253]. After neutral thermal hydrolysis, 2-hydroxyTHF **171** (Figure 8) was released as the major product from the treated DNA in vitro; *N*^2^-THF-dGuo **185** is considered the major precursor of this product [235]. The reduction of *N*^2^-THF-dGuo forms *N*^2^-(4-hydroxybutyl)-2′-deoxyguanosine **190** (*N*^2^-(4-HOB)-dGuo, Figure 12), which occurred in relatively low abundance compared to other types of NPYR DNA adducts in the liver DNA of rats. However, it remains one of the most abundant adducts among those formed by the oxonium ion **170** [247,252]. 

The other tetrahydrofuranyl DNA adducts *N*^6^-(tetrahydrofuran-2-yl)-2′-deoxyadenosine **184** (*N*^6^-THF-dAdo) and *N*^4^-(tetrahydrofuran-2-yl)-2′-deoxycytidine **186** (*N*^4^-THF-dCyd) were also identified in calf thymus DNA incubated with α-acetoxyNPYR. Two unstable Thd adducts, *O*^2^-(tetrahydrofuran-2-yl)thymidine **187** (*O*^2^-THF-Thd) and *O*^4^-(tetrahydrofuran-2-yl)thymidine **188** (*O*^4^-THF-Thd), were also formed and verified by their reduction products [255]. Some of the tetrahydrofuranyl DNA adducts were also observed in the reactions of oxidized THF with deoxyribonucleosides in vitro [256]. In a study in which F344 rats were gavaged with a single dose of NPYR and sacrificed after 16 h, *O*^2^-THF-Thd **187** (quantified in the reduced form **192**) predominated among the tetrahydrofuranyl DNA adducts at all 3 doses (see Table 1) [247]. In the chronic rat study with doses of NPYR administered in drinking water at 600 ppm for 1 week or 200 ppm for 4 or 13 weeks, dGuo and Thd adducts persisted in much higher abundance than the dAdo adduct. However, they did not seem to accumulate over the study course [252].

##### DNA Adducts Formed by NPYR-Derived Crotonaldehyde 

In the reactions of dGuo with two stable regiochemically activated compounds, α-acetoxyNPYR **165** and 4-(carbethoxynitrosamino)butanal **166** (Figure 8), two diasteromeric DNA adducts, (6*S*,8*S*)- and (6*R*,8*R*)-*N*1,*N*^2^-propanodeoxyguanosine **194** and **195** (*N*1,*N*^2^-propano-dGuo, Figure 12), were the major products, identical to the products formed by crotonaldehyde **177** [257]. Those two adducts were also formed by the reaction of α-acetoxyNPYR with dGuo, DNA, or RNA in vitro [244,245,249]. In the hepatic DNA of rats treated with NPYR in drinking water for 14 days, *N*1,*N*^2^-propano-dGuo (**194** and **195** together) occurred at the level of 0.06 μmol/mol Gua. This adduct was also formed in the skin DNA of mice topically treated with crotonaldehyde. However, the concentration of *N*1,*N*^2^-propano-dGuo (**194** and **195** together) in the rat liver DNA was significantly lower (~10,000-fold) than *N*7,8-butano-Gua **179** [251]. The diastereomer (6*S*,8*S*)-*N*1,*N*^2^-propano-dGuo **194** was preferentially formed over its (6*R*,8*R*)-counterpart **195** by crotonaldehyde [233]. 

In addition to the methyl-substituted *N*1,*N*^2^-propano-dGuo adducts, two new exocyclic 7,8-guanine adducts, *cis*- and *trans*-2-amino-7,8-dihydro-8-hydroxy-6-methyl-3*H*-pyrrolo [2,1-*f*]purine-4(6*H*)-one **196** and **197** (*N*7,8-Cro-Gua, Figure 12), were also identified in vitro in calf thymus DNA treated with α-acetoxyNPYR or crotonaldehyde but not in vivo in rat liver DNA [244,245]. They are likely formed via initial Michael addition followed by cyclization, the same mechanism for the selective formation of *N*1,*N*^2^-propano-dGuo [258]. The same two diasteromeric adducts **196** and **197** were also detected in crotonaldehyde-treated DNA in vitro and in vivo [244,251].

##### Other DNA Adducts Related to NPYR Metabolism

As depicted in Figure 8, the metabolite 3-hydroxybutanal **175** can dimerize to form paraldol **176**, which has been detected in the hydrolysates of DNA treated with α-acetoxyNPYR. Paraldol was also released from crotonaldehyde-treated DNA after neutral thermal hydrolysis but in much higher abundance, suggesting the presence of the unstable paraldol-releasing adduct(s) formed by α-acetoxyNPYR is likely via the intermediate crotonaldehyde [235]. Two paraldol-dGuo adducts were characterized in crotonaldehyde-treated DNA. They are *N*^2^-(2-(2-hydroxypropyl)-6-methyl-1,3-dioxan-4-yl)-2′-deoxyguanosine **198** (*N*^2^-paraldol-dGuo, Figure 13) and *N*^2^-(2-(2-hydroxypropyl)-6-methyl-1,3-dioxan-4-yl)deoxyguanylyl-(5′–3′)-thymidine **199** (*N*^2^-paraldol-dGuo-(5′–3′)-Thd) [233]. The major paraldol-releasing DNA adduct was later characterized to be *N*^2^-(3-hydroxybutylidene)-2′-deoxyguanosine **200**, a Schiff base that was unstable at the nucleoside level but appeared to be stable in DNA. The level of **200** exceeded the Michael addition products of tricyclic *N*1,*N*^2^-propano-dGuo adducts **194** and **195** in crotonaldehyde- and α-acetoxyNPYR-treated DNA [234,259].

In the reaction of α-acetoxyNPYR **165** with dGuo, *N*7-(*N*-nitrosopyrrolidin-2-yl)guanine **202** (Figure 13) was identified as the first example of a nitrosamine adduct retaining the *N*-nitroso moiety [259]. It was considered to be formed through the intermediacy of the nitrosiminium ion **201** [260]. A new cyclized adduct 2-(2-hydroxypyrrolidin-1-yl)-2′-deoxyinosine **203** (*N*^2^-Py(OH)-dI) was for the first time identified and confirmed by its synthetic standard in the reaction of α-acetoxyNPYR with both dGuo and DNA; it can be reduced to 2-(pyrrolidin-1-yl)-2′-deoxyinosine **204** (*N*^2^-Py-dI) [259].

#### 4.1.4. Mutagenicity and Genotoxicity of NPYR-Derived DNA Adducts

NPYR is mutagenic towards *E. coli* and *S. typhinurium* after hepatic microsomal activation [261,262]. A few DNA adducts formed by NPYR metabolism such as *N*1,*N*^2^-propano-dGuo can also arise from crotonaldehyde. The mutagenicity and genotoxicity of these types of DNA adducts have been extensively reviewed before [263,264,265]. However, the mutagenicity and genotoxicity of the other types of NPYR DNA adducts, such as **184**–**186** and their reduced forms **189**–**193**, still warrant investigation. 

#### 4.1.5. Human DNA Adducts Related to NPYR Metabolism

The *N*1,*N*^2^-propano-dGuo adducts have been detected in human tissues [266,267]. We have quantified (6*S*,8*S*)-*N*1,*N*^2^-propano-dGuo **194** and (6*R*,8*R*)-*N*1,*N*^2^-propano-dGuo **195** in human liver, lung, and white blood cells [268]. (6*S*,8*S*)-*N*1,*N*^2^-Propano-dGuo **194** occurred at a mean (± SD) concentration of 6.70 ± 2.92 and 7.19 ± 4.14 fmol/μmol dGuo, respectively, in liver and lung DNA. However, the detection rates were low; only 4 out of 23 liver samples and 16 out of 45 lung samples were positive. Similarly, (6*R*,8*R*)-*N*1,*N*^2^-propano-dGuo **195** occurred at mean (±SD) concentrations of 7.87 ± 4.47 and 12.8 ± 7.6 fmol/μmol dGuo, respectively, in the same liver and lung DNA. No statistically significant difference was observed between the occurrence of the two diastereomers. Neither of the two isomers was detected in the DNA of 11 human white blood cell samples [268]. However, considering the fact that the most abundant NPYR DNA adduct, *N*7,8-butano-Gua **179**, has not been detected in the human tissues, the origin of *N*1,*N*^2^-propano-dGuo is likely not due to the exposure of NPYR but rather to other pathways such as the endogenous formation of crotonaldehyde from lipid peroxidation [269]. Presently, there is no direct evidence for the presence of NPYR-DNA adducts specifically in human tissues. However, the studies on NPYR-DNA damage did lead to research on crotonaldehyde-DNA interactions and their detection in human tissue samples as noted above. 

### 4.2. N-Nitrosopiperidine (NPIP)

#### 4.2.1. Exposure and Carcinogenicity

NPIP (**9**, Figure 1) has been reported in water, spices, and foods such as cheese, smoked fish, and processed meat [73,270]. The average concentration of NPIP was 0.5 ± 0.1 ng/g in food and 7.9 ± 4.0 ng/L in potable water [9]. The formation of NPIP by the nitrosation of amine precursors such as piperidine has been associated with concentrations of sodium nitrite during the production of dry fermented sausages [271]. The presence of NPIP in tobacco products has been documented but is currently at extremely low levels [73,77].

The IARC reviewed the carcinogenicity data of NPIP in 1978 and reaffirmed its Group 2B classification in 1987. It has shown carcinogenic effects in multiple laboratory animals such as mice, rats, hamsters, and monkeys [73]. In mice, NPIP induced primarily liver and lung tumors when administered in the diet and caused lung adenomas when administered in the drinking water or by i.p. injection. In rats, esophageal and hepatic tumors were induced by NPIP administered in drinking water. However, the target organ shifted to the nasal cavity when administered by s.c. injection. Carcinomas of the esophagus and pharynx were mainly observed by i.v. injection of NPIP [73]. The ability of NPIP to cause esophageal tumors is striking when compared to its close analog NPYR, which never causes esophageal tumors in rats. This was confirmed by Gray et al. in a large 2-year study. The incidence of hepatic and esophageal tumors in rats was dose-responsive to the concentration of NPIP in the drinking water [220]. In hamsters, the trachea appears to be the primary target organ by NPIP after s.c. injection. In monkeys, hepatocellular carcinomas were observed after oral dosing with NPIP [73]. 

#### 4.2.2. Metabolism

NPIP is primarily biotransformed by hepatic P450s with the additional contribution of cytosolic proteins [272,273,274]. Catalyzed primarily by P450 2As [226,275,276], NPIP undergoes α-hydroxylation to form the unstable α-hydroxyNPIP **206** (Figure 9). This intermediate has been trapped in the form of α-hydroxyNPIP phosphate ester **221** under near-ultraviolet irradiation conditions [277]. After the decomposition of α-hydroxyNPIP and the spontaneous loss of H_2_O and N_2_, the electrophilic intermediate carbocation **213** is formed. The carbocation **213** can form the oxonium ion **216** via intramolecular cyclization or the isomeric carbocation **211** via a 1,2-*H* shift. Their solvolysis products, 5-hydroxypentanal **214** (in equilibrium with 2-hydroxytetrahydropyran **217** (THP-OH)) [278,279,280] and the reduced form 1,5-pentanediol **215 [272]** and 4-hydroxypentanal **212** (in equilibrium with 2-hydroxy-5-methyltetrahydrofuran **210**) [278] or the potential elimination/oxidation product 4-oxopent-2-enal **209** (acetylacrolein) [278], have been identified in support of the proposed mechanism of NPIP α-hydroxylation. It is noteworthy that 4-oxopent-2-enal **209** may also result from the metabolism of 2-methylfuran [281]. 

Beta- and γ-hydroxylation of NPIP have also been suggested as potential metabolic pathways (Figure 9). *N*-Nitroso-4-hydroxypiperidine **219** and *N*-nitroso-4-piperidone **220** were detected in vitro in early studies with rat liver microsomes [282,283]. *N*-Nitroso-3-hydroxypiperidine **218** and *N*-nitroso-4-hydroxypiperidine **219** were minor products compared to 5-hydroxypentanal **214** in studies with guinea pig liver microsomes [280]. 

The metabolic bioactivation pathways illustrated in Figure 9 were also supported by the structure-carcinogenicity studies. Methyl substitution at the α-carbons of NPIP significantly decreased tumor formation in rats administered NPIP or its methyl analogs in drinking water [284]. The β- or γ-substitutions (methyl and hydroxy) did not affect the carcinogenicity of NPIP in rats [284,285].

#### 4.2.3. DNA Adducts Formed by NPIP Metabolism

In the reaction mixture of α-acetoxy-*N*-nitrosopiperidine **205** (α-acetoxyNPIP, Figure 9) with dGuo, a peak corresponding to 7-(2-oxopropyl)-5,9-dihydro-9-oxo-3-β-D-deoxyribofuranosylimidazo [1,2-*a*]purine **222** (7-(2-oxopropyl)-*N*1,*N*^2^-etheno-dGuo, Figure 14) was observed. Its formation was proposed to occur via the intermediate 4-oxopent-2-enal **209** [254,278]. The unstable hemiaminal precursor for 7-(2-oxopropyl)-N*N*1,*N*^2^-etheno-dGuo has been identified in vitro, as such or in its reduced form. It was characterized as 7-(2-oxopropyl)-5-hydroxy-5,6,7,9-tetrahydro-9-oxo-3-β-D-deoxyribofuranosylimidazo [1,2-*a*]purine **223** [286].

A pair of two diastereomeric *N*^2^-(3,4,5,6-tetrahydro-2*H*-pyran-2-yl)-2′-deoxyguanosine **224** (*N*^2^-THP-dGuo, Figure 14) were identified as the major products in the same reaction mixture of dGuo compared to 7-(2-oxopropyl)-*N*1,*N*^2^-etheno-dGuo **222** [254]. Similarly, in the reaction mixture of α-acetoxyNPIP **205** with calf thymus DNA, *N*^2^-THP-dGuo was clearly formed, whereas 7-(2-oxopropyl)-*N*1,*N*^2^-etheno-dGuo was minimally observable. THP-OH **217** (Figure 9) was released in the DNA hydrolysate after neutral thermal hydrolysis, probably resulting from *N*^2^-THP-dGuo [253].

#### 4.2.4. Human DNA Adducts Related to NPIP Metabolism

In 2019, Totsuka et al. reported their DNA adductome investigation of esophageal cancer patients in China [287]. They found that a distinctive pattern of NPIP-derived DNA adduct *N*^2^-THP-dGuo **224** (Figure 14) formed in the tissues of esophageal cancer patients in the high-incidence area versus those in the low-incidence area. The difference in the occurrence of this adduct was also statistically significant (*p* < 0.01) in the peripheral blood samples of patients from the two areas. The exposure to NPIP was not likely due to smoking or drinking since all the samples were from patients who were nonsmokers and nondrinkers. Drinking water and vegetables were potential sources that have been preliminarily investigated [287].

NPIP induced preferentially AT to CG transversions in *gpt* delta transgenic rats. However, the predominant somatic mutation of esophageal tumor samples was CG to TA transitions; no distinctive mutational pattern was identified to associate with esophageal cancer in the high-incidence area. In the p53 genes of ~90% of patients, GC to AT transitions predominated, in agreement with the findings from their previous study in 2005 [288]. There was only a weak correlation observed between one of the mutational signatures (signature 17 in the COSMIC database) and the *N*^2^-THP-dGuo levels among 19 subjects (r = 0.44; *p* < 0.05). Taken together, this study suggests that NPIP-induced DNA adducts can at least partially contribute to esophageal carcinogenesis [287].

### 4.3. N-Nitrosomorpholine (NMOR)

#### 4.3.1. Exposure and Carcinogenicity

NMOR (**10**, Figure 1) has been detected in water, food, cosmetics, and occupational airspaces [289,290,291,292,293,294,295]. It has also been documented to occur in smokeless tobacco products but at very low concentrations [77]. Similar to other *N*-nitrosamines, NMOR can be formed easily by nitrosation of its parent amine morpholine with NaNO_2_. It is one of the most rapidly formed nitrosamine products [296]. When morpholine was administered in combination with nitrite to rats by gavage, the formation of NMOR was readily measured in the urine, with the extent of morpholine nitrosation being 0.5–12% depending on the doses [297]. Mice exposed to ^15^NO_2_ and administered morpholine by gavage also formed NMOR with the highest concentrations detected in the skin followed by the stomach [298]. Human exposure to morpholine and its analogs has been considered substantial from food and drugs [270,299,300], tobacco usage [301], and rubber and tire manufacturing [302,303]. Thus, the in vivo formation of NMOR has been considered likely to occur in humans. Low levels of NMOR have been detected in human urine and gastric juice [304,305,306].

The carcinogenicity data of NMOR was reviewed by IARC in 1978 and 1987, and it was classified as a group 2B carcinogen. NMOR primarily induces liver, bile duct, and kidney tumors in mice and rats when given in the drinking water or by i.v. injection; it causes tumors of the respiratory system (trachea and nasal cavity) of hamsters by s.c. injection [73]. After 1978, there were some new carcinogenicity data reported. When NMOR was administered in the drinking water to A/J mice, lung tumors occurred at a relatively low total dose of 53–55 μmol/mouse. It induced 100% liver tumor incidence in F344 rats at a high total dose of 1.1 mmol/rat [190]. It is interesting to note that 2-/6-methylation of NMOR shifts its organospecificity. In Syrian golden hamsters treated by gavage, NMOR induced primarily nasal cavity tumors whereas *N*-nitroso-2-methylmorpholine induced tumors in the nasal cavity and liver [307]; *N*-nitroso-2,6-dimethylmorpholine caused liver and pancreas tumors [308]. In rats, NMOR is a potent liver carcinogen whereas *N*-nitroso-2,6-dimethylmorpholine is carcinogenic to the esophagus and nasal cavity [7,309].

#### 4.3.2. Metabolism

After a single i.p. injection, NMOR was rapidly distributed throughout the rat tissues and metabolized to a relatively high extent with 24% of the dose excreted unchanged in the urine after 18 h [310]. It is metabolized via both α- and β-hydroxylation pathways (Figure 10). 

Hydroxylation on the 2-carbons of NMOR appears to be the major biotransformation pathway, since NHEG **135**, derived from the β-hydroxylation product NHMOR **137** via the intermediate **136**, has been detected predominantly among the rat urinary metabolites, accounting for 33% or 37% of total metabolites [311]. It has been used as a reliable biomarker to monitor the in vivo formation of NMOR [297]. However, NHEG can also be formed by NDELA and *N*,*N*-dinitrosopiperazine, although to the lesser extent of <10% and 22% from each compound, respectively [184,312]. It is noteworthy that the group 2B carcinogen NDELA **6** is also formed by the reduction of intermediate **136** during NMOR metabolism, accounting for 12% of total urinary metabolites [311,313]. NDELA can be reversibly oxidized to form the same products **135**, **136**, and **137** (see Figure 6) as in the case of NMOR. 

When the hydroxylation occurs on the 3-carbon of NMOR, the unstable product *N*-nitroso-3-hydroxymorpholine **228** (α-hydroxyNMOR) is formed. By analogy to NPIP metabolism, a potential α-hydroxyNMOR acetate ester **234** has been suggested to be similarly formed under near-ultraviolet light irradiation conditions [314]. α-HydroxyNMOR decomposes quickly to form the diazohydroxide **229**, which can also result from the regiochemically activated precursors α-acetoxy-*N*-nitrosomorpholine **225**, *N*-nitrosomorpholine hydroperoxide **226**, and (2-(carbethoxynitrosamino)ethoxy)ethanal **227** [315,316,317]. The diazohydroxide spontaneously loses one molecule of H_2_O and N_2_ and forms the carbocation **231** and its isomeric forms, **232** and **233**. The carbocation **231** forms the solvolysis product, 2-(2-hydroxyethoxy)acetaldehyde **235,** in equilibrium with 2-hydroxy-1,4-dioxane **236**; the carbocation **232** forms acetaldehyde **42** and glycolaldehyde **130** after solvolysis; the other carbocation **233** forms glyoxal **133** [311,318]. The major ultimate product arising from NMOR α-hydroxylation was (2-hydroxyethoxy)acetic acid **237** in rat urine. A significant isotope effect was observed in the formation of **237** when the α-carbons of NMOR were deuterated (16% vs. 3.4%) [311]. It has been shown that 1,4-dioxan-2-one **238** exists in aqueous solutions in equilibrium with **237** with an equilibrium constant *K*_OA_ of 0.034 ± 0.002 M. It appears to be a common non-carcinogenic metabolite of the carcinogens NMOR, dioxane, and diethylene glycol [319].

The β-hydroxylation product of NMOR—NHMOR **137** (Figure 10)—is highly mutagenic [189,320,321] and forms DNA adducts in vitro [315]. It is, however, non-tumorigenic or marginally tumorigenic to laboratory animals [190]. To explain the apparent lack of carcinogenicity of NHMOR, Loeppky et al. investigated the metabolic profiles of two NHMOR α-hydroxylation metabolites—*N*-nitroso-2,3-dihydroxymorpholine **139** and *N*-nitroso-2,5-dihydroxymorpholine **142**—using their stable precursors, *N*-nitroso-3-acetoxy-2-hydroxymorpholine **138** and *N*-nitroso-5-acetoxy-2-hydroxymorpholine **141**. The hydrolytic decomposition products of **139** were glyoxal **133** (95%), ethylene glycol **129** (55%), acetaldehyde **42** (10%), and acetic acid; the hydrolytic decomposition products of **142** were 2-acetoxyacetaldehyde **144** (65%), glycol aldehyde **130** (15%), glyoxal **133** (trace), and acetic acid. The high yield of 2-acetoxyacetaldehyde **144** may be responsible for the low carcinogenicity of NHMOR if it is primarily hydroxylated at the 2-carbons [195].

#### 4.3.3. DNA Adducts Formed by NMOR Metabolism

A study by Stewart et al. in 1974 using [^14^C]NMOR suggested that six radioactive DNA adducts are formed by NMOR metabolism. One of those compounds was putatively identified to be *N*7-HOEt-Gua **69** (Figure 5) [310]. Fishbein et al. incubated nucleosides and calf thymus DNA with NMOR hydroperoxide **226** (Figure 10) and identified a panel of 2-ethoxyacetaldehyde purine adducts as shown in Figure 15. These include *N*3-, *N*^6^-, and *N*7-(2-oxoethoxyethyl)adenine (**245**, **246,** and **248**) and *N*1-, *N*^2^-, *O*^6^-, and *N*7-(2-oxoethoxyethyl)guanine (**249**, **250**, **252** and **253**), among which *N*7-(2-oxoethoxyethyl)guanine **253** and *O*^6^-(2-oxoethoxyethyl)guanine **252** are the two most abundant and *N*3-(2-oxoethoxyethyl)adenine **245** is the third, however, in much lower concentrations. Those 2-oxoethoxyethyl adducts slowly decayed to the corresponding hydroxyethyl adducts (**62**–**67** and **69**, Figure 5) with a faster rate in duplex DNA than in nucleosides. Interestingly, the two 2-oxoethoxyethyl adducts **246** and **250,** occurring at the exocyclic amino groups, also formed the corresponding intramolecular ring closure products **247** and **251,** respectively, during chemical standard synthesis. However, **247** and **251** were not detected in the treated nucleosides or calf thymus DNA [317]. The same group also identified the first cross-link DNA adduct 6-(2-(2-((9*H*-purin-6-yl)amino)ethoxy)ethoxy)-9*H*-purin-2-amine **254** (Figure 15) derived from α-hydroxyNMOR upon reduction. It was proposed to be formed by the carbocation **231** (Figure 10) attacking the *O*^6^-OH of guanine first, followed by the pendant aldehyde reacting with *N*^6^-NH_2_ of adenine [322].

We conducted an in vitro study of [2-(carbethoxynitrosamino)ethoxy]ethanal **227** (Figure 10) reacting with dGuo and demonstrated the formation of *N*1,*N*^2^-glyoxal-dGuo **145** (Figure 11) probably via glyoxal resulting from NMOR α-hydroxylation [309]. In the reaction mixture of NHMOR with dGuo, adduct **145** was also formed. The proposed mechanism of the formation of this adduct by NHMOR is depicted in Figure 11 [315]. 

Consistent with their metabolic profiles, the two precursors of α-hydroxyNHMOR **138** and **141** (Figure 10) showed different reactivity toward dGuo. In the in vitro incubation mixture with dGuo, compound **141** was less reactive compared to **138**. The dGuo adduct formed by **141** was *N*1,*N*^2^-etheno-dGuo **146** (Figure 11). The dGuo adduct primarily formed by **138** was *N*1,*N*^2^-glyoxal-dGuo **145** (Figure 11) along with two minor adducts, *N*7-HOEt-Gua **69** and *O*^6^-HOEt-Gua **67** (Figure 5) [195].

## 5. Concluding Remarks

The structural simplicity of the powerful carcinogen NDMA provided a unique opportunity to understand the role of metabolism and DNA adduct formation in carcinogenesis. This research has clearly demonstrated the main requisite steps in tumor formation by *N*-nitrosamines and other related organic carcinogens: formation of electrophilic intermediates catalyzed by cytochrome P450 enzymes, binding of those intermediates to sites in DNA, the role of DNA repair systems, mutagenesis resulting from certain unrepaired DNA adducts such as *O*^6^-Me-Gua, and consequent permanent mutations in critical growth control genes. This sequence of events is also quite well-established for NDEA and NMEA and forms the basis for the investigation of more structurally complex *N*-nitrosamines as described here. For these other *N*-nitrosamines, there are still gaps in our understanding of their metabolic activation pathways and the DNA adduct structures, repair mechanisms, and mutagenic properties of the persistent adducts. This review has presented an overview of the current status of this research.

Studies of the metabolic activation and DNA adduct formation of carcinogenic *N*-nitrosamines have long been part of the critical mass of research on these compounds. The focus of many cancer researchers on *N*-nitrosamines produced an overall *heightened awareness* of their ease of formation, levels of contamination in consumer products, and carcinogenic effects among other topics. This heightened awareness led to significant decreases in human *N*-nitrosamine exposures from the 1970s to the present. However, constant vigilance is necessary to maintain these lower exposure levels. The recent manufacturing errors leading to *N*-nitrosamine contamination of drugs would have been unimaginable in the 1980s and 1990s when the ease of *N*-nitrosamine formation under certain conditions was prominent in the eyes of chemists and toxicologists. 

There are approximately 200 different *N*-nitrosamines that have been documented to be carcinogenic in more than 30 animal species [323]. Humans exposed to these compounds are, no doubt, susceptible to their carcinogenic effects. The continuous evaluation of potential *N*-nitrosamine contamination such as *N*-nitrosomethyl-*n*-alkylamines in medicines is in progress by other groups [324]. Among those many *N*-nitrosamines, the compounds listed in Figure 1 comprise the most commonly detected carcinogenic *N*-nitrosamines from human daily exposure, occurring in food, water, drugs, and cosmetics. An understanding of the metabolism of these carcinogens and their mechanisms of DNA interactions is necessary basic knowledge for evaluating and controlling their potential carcinogenic effects in humans, and some metabolites and/or DNA adducts that are specific to carcinogen exposure in humans may serve as biomarkers for cancer etiology studies. 

With advances in bioanalytical methods, especially in the development of mass spectrometry, metabolism studies have evolved from radioisotope-labeling technology to high-resolution mass spectrometry. Furthermore, mass spectrometry-based metabolomics studies may provide rich chemical information on structures of potential metabolites of interest [325]. Similarly, DNA adductomics [326] is also emerging as a useful tool to investigate DNA adducts that are potential biomarkers for cancer etiology studies, as illustrated in the NPIP section [287]. 

In summary, the recent occurrence of NDMA and NDEA in some batches of FDA-approved drugs has heightened global awareness of the carcinogenic effects of *N*-nitrosamines, which comprise a significant number of carcinogens to which humans are exposed on a daily basis through food, water, drugs, cosmetics, and tobacco products. In this review, we provide a comprehensive and updated review of 10 *N*-nitrosamine carcinogens with a focus on their mechanisms of metabolic activation and DNA interactions. A better understanding of the metabolism and DNA adduct formation of *N*-nitrosamines can hopefully provide clues for relevant cancer etiology and prevention studies.

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
