# Peer review of "Metabolic Activation and DNA Interactions of Carcinogenic N-Nitrosamines to Which Humans Are Commonly Exposed"

_ijms, 2022, doi:10.3390/ijms23094559_

Round 1
Reviewer 1 Report
This is a timely and comprehensive review of information on the metabolism, carcinogenicity, and DNA adduct formation associated with 10 N-nitrosamine compounds humans are commonly exposed to. It is a very useful review and will serve as an excellent reference for these 10 chemicals. The numerous figures and metabolic schemes are especially valuable syntheses of current knowledge.
Minor comments and suggestions:
- 15, line 568. Consider revising the title of Figure 9 to more clearly indicate that the structures shown are analogues and/or metabolites of NDPA which form DNA adducts.
- 24, Table. Consider adding footnotes to the column subheadings for the 4-hydroxybutyl adducts, to indicate that detections of these adducts are in some cases the result of chemical reduction of the corresponding tetrahydrofuranyl DNA adducts – as discussed on p. 26 (lines 846-949).
- 31, line 1073. Suggest inserting the word ‘intermediate’ before 136.
- Consider mentioning that while this review focuses on 10 specific nitrosamine compounds, there are other nitrosamines that humans are likely exposed to. For example, thirteen N-nitrosomethyl-n-alkylamines, of which NMDA is the smallest/simplest member of the class, have been studied in carcinogenesis assays in laboratory animals, and been shown to induce tumors. These compounds are recognized as carcinogens in California, and the possibility of human exposure e.g., from contamination of drugs, raises concerns (Li et al 2021 Int J Environ Res Public Health Sep 8; 18(18):9465. Doi:10.33390/ijerpj18189465).
Reviewer 2 Report
Metabolic Activation and DNA Interactions of Carcinogenic N-Nitrosamines to Which Humans are Commonly Exposed
In this review, Li Y. and Hecht S.S. have evaluated the role of 10 carcinogenic N-nitrosamines and their metabolic activation and interaction with DNA.
This exhaustive paper comprises researches conducted within the last six decades on N-nitrosamines to which humans are frequently exposed, and it accurately, yet comprehensively, descibe the DNA adducts formation from the metabolism of the investigated compounds and their role in carcinogenesis.
In my opinion, the paper is well written and the schematic arrangement of the content enables a clear examination of the manuscript.
